# Vector-like quarks: status and new directions at the LHC

Avik Banerjee[a,b,1], Elin Bergeaas Kuutmann[c,2], Venugopal Ellajosyula[c,d,e,3],
Rikard Enberg[c,4], Gabriele Ferretti[b,5], Luca Panizzi[f,g,6]

[a] *Department of Theoretical Physics, Tata Institute of Fundamental Research, Homi Bhabha Road, Mumbai 400005, India*

[b] *Department of Physics, Chalmers University of Technology, Fysikgården, 41296 Göteborg, Sweden*

[c] *Department of Physics and Astronomy, Uppsala University, Box 516, SE-751 20 Uppsala, Sweden*

[d] *Department of Physics, Università di Genova, Via Dodecaneso 33, Genova, 16146 Italy*

[e] *INFN-Genova, Via Dodecaneso 33, 16146 Genova, Italy*

[f] *Department of Physics, Università della Calabria, Via P. Bucci, Cubo 31 C, I-87036 Cosenza, Italy*

[g] *INFN-Cosenza, I-87036 Arcavacata di Rende, Cosenza, Italy.*

## ABSTRACT

Experimental searches for vector-like quarks have until now only considered their decays into Standard Model particles. However, various new physics scenarios predict additional scalars, so that these vector-like quarks can decay to new channels. These new channels reduce the branching ratios into Standard Model final states, significantly affecting current mass bounds. In this article, we quantitatively assess the relevance and observability of single and pair production processes of vector-like quarks, followed by decays into both standard and exotic final states. We highlight the importance of large widths and the relative interaction strengths with Standard Model particles and new scalars. Then, we review the post-Moriond 2024 status of these models in light of available LHC data and discuss potential future strategies to enhance the scope of vector-like quark searches.

[1] avik.banerjee_205@tifr.res.in
[2] elin.bergeaas.kuutmann@physics.uu.se
[3] venugopal.e@protonmail.com
[4] rikard.enberg@physics.uu.se
[5] ferretti@chalmers.se
[6] luca.panizzi@unical.it

# 1 Introduction

After more than a decade of intense experimental activities at the Large Hadron Collider (LHC), no significant deviation from the predictions of the Standard Model (SM) of particle physics has been observed. Many comprehensive reviews of the current situation and planning reports for future activities have been put forward lately, see e.g. [1–5]. These general works focus on a broad category of excursions beyond the SM (BSM). One of the purposes of this article is to draw attention to a class of models that, while still well motivated, are "less minimal" in the sense of containing a richer spectrum of BSM particles. On the one hand, the presence of additional particles modifies the constraints from current experimental searches; on the other hand, it opens up new channels for future searches for new physics.

One class of models aiming at addressing the well-known problems with the SM entails the existence of vector-like fermions (VLF), commonly referred to as "partners" of SM fermions when they share the same electric charge. They were initially proposed in [6] in the context of technicolour, but since then their use has widened in scope [7]. These vector-like BSM fermions possess Dirac masses even in the absence of a vacuum expectation value of the Higgs boson, since their left-handed and right-handed chiralities transform identically under the SM gauge group. Due to this fact, they are also safe from gauge anomalies, in contrast to chiral fermions.

VLFs, particularly vector-like *quarks* (VLQs)[1], arise in composite Higgs models [9, 10] (CHM) where they provide the necessary interactions to misalign the vacuum and break the electroweak symmetry. The mixing between the VLQs and the top quark via partial compositeness interactions provides an explanation for the large top quark mass in the CHMs [6, 11–13]. VLQs can also appear in other theories, e.g., two-Higgs doublet models (2HDMs) [14–19].

Composite Higgs models are based on the breaking of a global symmetry $G$ of a strongly interacting sector, down to a symmetry $H$ with resulting coset space $G/H$, where the pseudo-Nambu–Goldstone bosons (pNGBs) of the symmetry breaking live [9]. The *minimal* CHM [10] is based on the coset $G/H = SO(5)/SO(4)$, and does not give rise to any additional light scalar other than the Higgs boson. All other composite Higgs models have larger cosets, and therefore additional scalar states.

When considering dynamical symmetry breaking arising from underlying four-dimensional confining theories, it is more natural to consider coset spaces $G/H$ where $G$ only contains $SU$ factors [20–22]. With this additional assumption, the minimal cases that also preserve custodial symmetry are indicated in Tab. 1. Note that the first coset $SU(4)/Sp(4) \cong SO(6)/SO(5)$ is usually referred to as the next-to-minimal CHM [23]. The various possible pNGBs and VLQs present in these models[2] are also listed in Tab. 1.

Experimental searches at LHC have focused on single or pair production of VLQs, followed by their decay into SM states. Most prominent have been searches for top-partners $T$ into the channels $Zt$, $Ht$, and $Wb$ and similar searches for bottom-partners $B$. No evidence for these particles has been found and this has led to lower bounds on their masses above the TeV, as we review below. The potential of the high-luminosity LHC (HL-LHC) and high-energy LHC (HE-LHC) to extend the reach is discussed in e.g. [5].

However, it is important to note that in these non-minimal scenarios the VLQs can also decay into BSM scalars. Bounds from the direct production of these scalars can be obtained, as discussed in, e.g. [26, 27].

Ref. [18] has considered what is arguably the simplest exotic channel, $T \to S^0 t$ ($S^0$ being a BSM neutral scalar), with subsequent decay $S^0 \to \gamma\gamma$ or $S^0 \to Z\gamma$. A more complete model leading to similar signature and with very suppressed $T$ decay into SM particles has been considered in [28]. In [29] a broader perspective was taken, including partners with exotic charges and (multi-)charged scalars. Additional exotic signatures are discussed by other groups in [8, 30–48].

Recent reviews from the ATLAS and CMS collaborations have summarised the results from VLQ searches during Run 2 in [49, 50]. In this study, we provide a more concise summary of these results using two summary plots: one for pair production (Fig. 6) and one for single production (Fig. 7) of VLQs, and compare the exclusion limits from both the ATLAS and CMS. We conduct a quantitative analysis to assess the relevance of pair production vis-a-vis single production and the significance of exotic decay channels of VLQs in future searches. Additionally, we highlight several novel aspects that could help designing future strategies for studying VLQs.

In this work we will not delve into the details of the theoretical constructions of CHMs, which are reviewed elsewhere (e.g. [28]) and will simply define the scope of the BSM models of interest. We only consider simplified models — theories where the SM particle content is extended by adding one vector-like quark $\Psi$ and one scalar multiplet $S$ with definite $SU(3)_c \times SU(2)_L \times U(1)_Y$ quantum numbers, see Tab. 1. We further assume that only the SM Higgs doublet can receive a vacuum expectation value ($v$), to avoid strong constraints from electroweak

---

[1]In this study, we will primarily focus on the $SU(3)_c$ colour triplet VLQs, and colourless electroweak scalars. See e.g. [8], for a discussion on VLFs in higher $SU(3)_c$ representations.

[2]An additional advantage of these constructions is that they can be studied on the lattice, see [24] and [25] for recent developments.

| Spin | Quantum numbers $[SU(3)_c \times SU(2)_L]_{U(1)_Y}$ | Components | Composite Higgs models | | |
|---|---|---|---|---|---|
| | | | $\frac{SU(4)}{Sp(4)}$ | $\frac{SU(5)}{SO(5)}$ | $\frac{SU(4)^2}{SU(4)}$ |
| 0 | $(\mathbf{1},\mathbf{1})_0$ | $S_{\mathbf{1}}^0$ | ✓ | ✓ | ✓ |
| | $(\mathbf{1},\mathbf{2})_{1/2}$ | $\left(S_{\mathbf{2}}^+, S_{\mathbf{2}}^0\right)$ | – | – | ✓ |
| | $(\mathbf{1},\mathbf{3})_0$ | $\left(S_{\mathbf{3}}^+, S_{\mathbf{3}}^0, S_{\mathbf{3}}^-\right)$ | – | ✓ | ✓ |
| | $(\mathbf{1},\mathbf{3})_1$ | $\left(S_{\mathbf{3}}^{++}, S_{\mathbf{3}}^+, S_{\mathbf{3}}^0\right)$ | – | ✓ | – |
| 1/2 | $(\mathbf{3},\mathbf{1})_{2/3}$ | $T_{2/3}$ | | | |
| | $(\mathbf{3},\mathbf{1})_{-1/3}$ | $B_{-1/3}$ | VLQs arise as | | |
| | $(\mathbf{3},\mathbf{2})_{1/6}$ | $(T_{2/3}, B_{-1/3})$ | irreducible | | |
| | $(\mathbf{3},\mathbf{2})_{7/6}$ | $(X_{5/3}, T_{2/3})$ | representations | | |
| | $(\mathbf{3},\mathbf{3})_{-1/3}$ | $(T_{2/3}, B_{-1/3}, Y_{-4/3})$ | of the unbroken | | |
| | $(\mathbf{3},\mathbf{3})_{2/3}$ | $(X_{5/3}, T_{2/3}, B_{-1/3})$ | subgroups $Sp(4)$, | | |
| | $(\mathbf{3},\mathbf{3})_{5/3}$ | $(\tilde{Y}_{8/3}, X_{5/3}, T_{2/3})$ | $SO(5)$, and $SU(4)$ | | |

**Table 1:** List of scalars arising in CHMs, with the specified coset spaces, together with the most common VLQ representations for these cosets. We do not list the SM Higgs doublet, with quantum numbers $(\mathbf{1},\mathbf{2})_{1/2}$, which arises in all cosets.

precision observables.

We thus consider an effective Lagrangian including one VLQ $\Psi$ and one scalar $S$ with definite quantum numbers under the SM gauge group,

$$\mathcal{L} = \mathcal{L}_{SM} + \mathcal{L}_{NP}^{\leq 4} + \mathcal{L}_{NP}^5 + \dots, \tag{1}$$

where $\mathcal{L}_{NP}^{\leq 4}$ denotes the new physics Lagrangian up to dimension-four operators, and $\mathcal{L}_{NP}^5$ denotes the Lagrangian containing only dimension-five operators. Schematically, $\mathcal{L}_{NP}^{\leq 4}$ is given by

$$\mathcal{L}_{NP}^{\leq 4} \supset \bar{\Psi}(i\not{D} - m_\Psi)\Psi + |D_\mu S|^2 - V(S, H) + \mu\bar{\Psi}f + y_H\bar{\Psi}fH + y_S\bar{\Psi}fS + \tilde{y}_S\bar{\Psi}\Psi S + \tilde{y}_f\bar{f}'fS, \tag{2}$$

where $f, f'$ denote SM fermions. Note that not all terms might be present depending on the representations of $\Psi$ and $S$. The $\mu$, $y_H$ and $y_S$ terms in (2) lead to partial compositeness interactions. A list of operators with at least one VLQ that contribute to $\mathcal{L}_{NP}^5$ is shown in Tab. 2.

| $\Psi$ + SM fields | $\bar{\Psi}\sigma^{\mu\nu}\Psi X_{\mu\nu}$ | $\bar{\Psi}\sigma^{\mu\nu}f X_{\mu\nu}$ | $\bar{\Psi}f H^2$ | $\bar{\Psi}\Psi H^2$ |
|---|---|---|---|---|
| $\Psi$ + $S$ + SM fields | | $\bar{\Psi}f HS$ | $\bar{\Psi}f S^2$ | $\bar{\Psi}\Psi S^2$ |

**Table 2:** Dimension-5 operators involving at least one VLQ. $X_{\mu\nu}$ denotes any of the SM gauge boson field strengths.

In Sec. 2 we discuss the production modes of the VLQs at the LHC, specifically focusing on the issue of if, and when, single VLQ production dominates over pair production. In Sec. 3, we present a quantitative estimate for the relevance of exotic decays of VLQs through a simple example. The current experimental situation is detailed in Sec. 4, where the main results are summarised in two plots: Fig. 6 and Fig. 7. Prospects for VLQ searches at Run 3 of LHC and HL-LHC are discussed in Sec. 5. In the Appendix A we list all Run 2 ATLAS and CMS searches relevant for these models. In Appendix B we present a list of final states which can be studied to search for a vector-like top partner.

## 2    VLQ pair or single production?

Assuming that VLQ interactions with SM fermions exclusively involve SM quarks of the third generation, VLQs can be produced at the LHC either in pairs, or singly in association with a third generation quark and a light jet. Relevant Feynman diagrams are shown in Fig. 1.

The cross-section for pair-production of VLQs is driven by QCD interactions, thus in the narrow-width approximation (NWA), the cross-section for QCD pair production depends exclusively on the VLQ mass. The cross-section for single production, on the other hand, is also proportional to the couplings of the VLQs to electroweak gauge bosons and third generation quarks, hereafter referred to as the EW couplings of VLQs. These additional couplings are model-dependent free parameters which can span a large range of values and thus determine the relevance of single production channels relative to pair production.

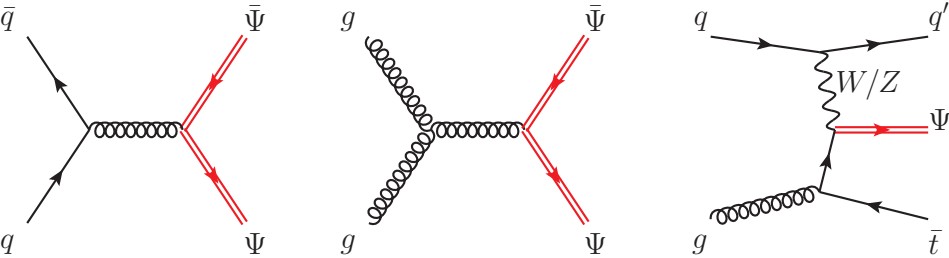

**Figure 1:** Representative Feynman diagrams contributing to pair and single production of the VLQ $\Psi$ at hadron collider.

Single production of VLQ is expected to become dominant with respect to pair production for growing VLQ mass, due to the smaller phase space suppression and greater contribution of quark PDFs with respect to gluon PDFs at higher energy scales. If the couplings of the VLQ to SM bosons are too small, single production might be suppressed for any VLQ mass within the energy reach of the LHC; if on the other hand they are sufficiently large such that single production can be observed at the LHC, the total width of the VLQ may become large enough so that NWA is no longer accurate [51].

While experimental searches have already considered large width in single production [52–57], it is essential to also consistently take into account large width effects in pair production [58]. This is due to the increasing significance of off-shell contributions, which are sub-dominant in the NWA. Moreover, non-factorisable contributions where the VLQ propagates only in $t$-channel topologies, and interferences with the SM irreducible background, both of which are absent by definition in the NWA, also gain importance for large width.

Hence, to properly simulate signals associated with VLQs having large width one should include the decay products of the VLQs in the final state, instead of assuming on-shell propagation of the VLQs. The cross-section associated to the new physics signal is a weighted sum of the various processes corresponding to the final states which can be reached by the propagation of the VLQ, as shown below

$$\sigma_{\text{pair}}(m_\Psi, \Gamma_\Psi) = \sum_{a,b} \kappa_a^2 \kappa_b^2 \, \hat{\sigma}_{pp \to a\bar{b}}(m_\Psi, \Gamma_\Psi) + \sum_{a,b} \kappa_a \kappa_b \, \hat{\sigma}_{pp \to a\bar{b}}^{int}(m_\Psi, \Gamma_\Psi) \, , \tag{3}$$

$$\sigma_{\text{single}}(m_\Psi, \Gamma_\Psi) = \sum_{a,q} \kappa_a^2 \kappa_q^2 \, \hat{\sigma}_{pp \to a\bar{q}j}(m_\Psi, \Gamma_\Psi) + \sum_{a,q} \kappa_a \kappa_q \, \hat{\sigma}_{pp \to a\bar{q}j}^{int}(m_\Psi, \Gamma_\Psi) + c.c. \, , \tag{4}$$

where $a, b$ are 2-body SM final states, e.g. $Wb$, $Zt$ or $Ht$ for a top partner, and their conjugates, while $\kappa_{a,b}$ denote the strength of the three point vertices involving $\Psi$ and the corresponding SM particles. Note that the above signal parametrisation, which uses a weighted sum of independent elements to account for subtle effects such as interferences and finite widths, allows to construct a complete set of simulated Monte Carlo samples of the signal. This can be used to describe the kinematical properties of final states for any new-physics benchmark, characterised by specific values of masses, couplings and total widths, therefore reducing the number of dedicated numerical simulations. The samples can also be used to determine the experimental efficiencies of any search, generalising this treatment also for future analyses.

For single production the $q$ summation is related to the coupling between $\Psi$ and the third-generation SM quark $q$ produced in association via interactions with $W$ or $Z$. The reduced cross-sections $\hat{\sigma}$, which depend only on the relevant kinematical parameters, mass $m_\Psi$ and total width $\Gamma_\Psi$, have been factorised from the couplings, and the interference contribution $\hat{\sigma}^{int}$ with purely SM processes leading to the same final states have been taken into account. For single $\Psi$ production, the contribution of the charge-conjugate process producing $\bar{\Psi}$ is considered as well.

As an example, let us consider a well-studied minimal extension of the SM with a vector-like top partner $T$ and focus on its interactions with the SM (the addition of interactions with new scalars will be studied in Sec. 3). We further assume that $T$ is either a $SU(2)_L$ singlet or a doublet with the following branching ratios [7][3]

$$BR_{T \to Wb} : BR_{T \to Zt} : BR_{T \to Ht} = 2 : 1 : 1, \quad \text{for singlet,}$$
$$BR_{T \to Wb} : BR_{T \to Zt} : BR_{T \to Ht} = 0 : 1 : 1, \quad \text{for doublet,}$$

and perform a comparative study between the cross-sections of signal and background for varying width/mass $(\Gamma_T/m_T)$ ratios. Note that the doublet $T$ can reside in either a $(T, B)$ or a $(X, T)$ doublet, see Tab. 1. The branching ratio of a $(T, B)$ doublet depends on its couplings with the SM right-handed quarks, we consider the branching ratio pattern when the Yukawa coupling with $b_R$ is zero. Even if BSM decays are not explicitly

---

[3]These relations are valid in the asymptotic limit of $m_T \gg m_i$, with $i$ being any SM boson $T$ can decay to, where the Goldstone boson equivalence theorem holds. We will consider this approximation valid for the entire range of $T$ masses explored here.

considered in this example, typically additional new physics, for example in the form of other VLQs or scalars with varying charges leading to compensations of loop effects, is necessary to evade constraints from flavour or EW precision observables [58–61].

The cross-sections of the irreducible SM backgrounds corresponding to all the combinations of possible final states for pair and single production are provided in Tab. 3.[4] Let $\sigma_{S+B}$ be the signal plus background cross-

| SM irr. bkg. for $T\bar{T}$ final states | | SM irr. bkg. for single $T$ final states | |
|---|---|---|---|
| Final state | Cross-section (pb) | Final state | Cross-section (pb) |
| $W^+bW^-\bar{b}$ | $500^{+28.7\%+5.3\%}_{-20.9\%-5.3\%}$ | $W^\pm b\bar{b}j$ | $114^{+12.9\%+1.5\%}_{-10.9\%-1.5\%}$ |
| $W^+bZ\bar{t} + W^-\bar{b}Zt$ | $1.140^{+29.3\%+4.8\%}_{-21.1\%-4.8\%}$ | $W^+b\bar{t}j + W^-\bar{b}tj$ | $8.17^{+16.1\%+2.1\%}_{-13.1\%-2.1\%}$ |
| $W^+bH\bar{t} + W^-\bar{b}Ht$ | $0.753^{+28.7\%+5.12\%}_{-20.9\%-5.12\%}$ | $Zt\bar{b}j + Z\bar{t}bj$ | $0.35^{+18\%+1.9\%}_{-14.3\%-1.9\%}$ |
| $ZtZ\bar{t}$ | $0.0015^{+27.5\%+3.5\%}_{-20.2\%-3.5\%}$ | $Ht\bar{b}j + H\bar{t}bj$ | $0.034^{+18,7\%+2.2\%}_{-14.7\%-2.2\%}$ |
| $ZtH\bar{t} + Z\bar{t}Ht$ | $0.0013^{+28.2\%+3.7\%}_{-20.5\%-3.7\%}$ | $Zt\bar{t}j$ | $0.0084^{+18\%+1.9\%}_{-14.3\%-1.9\%}$ |
| $HtH\bar{t}$ | $0.0007^{+28.3\%+4.3\%}_{-20.6\%-4.3\%}$ | $Ht\bar{t}j$ | $0.0041^{+18\%+2.6\%}_{-14.2\%-2.6\%}$ |

**Table 3:** Cross-sections at 13.6 TeV, in decreasing order, for irreducible backgrounds corresponding to final states compatible with pair or single production of a vector-like top partner $T$. The uncertainties correspond to scale and PDF systematics respectively.

section for a certain process, including the interference effects between signal and background, while $\sigma_B$ be the background-only cross-section for the same process. The relative difference between the two cross-sections, denoted by $\delta_{SB} \equiv (\sigma_{S+B} - \sigma_B)/\sigma_B$, are shown in Fig. 2, for each of the final states and for two different widths $\Gamma_T/m_T = 0.01$ and $0.1$. In the lower part of the plots the relative ratios between the cross-sections computed with a proper finite width approach and those in the NWA are also shown.

The effects of finite width are more pronounced for processes which have a low background, *i.e.* those involving one or more top quarks in the final state. The relevance of single production with respect to pair production increases with the VLQ width, especially for $m_t \lesssim 2$ TeV where cross-sections are higher, as the single production cross-section depends on the same couplings which determine the width. Above 2 TeV, however, interference effects become relatively important for both single and pair production processes, in some cases leading to mildly negative relative differences $\delta_{SB}$. This would potentially result in a *deficit* of events.

As expected, the deviations with respect to the NWA significantly increase with the width. However, interestingly, even in the case of a reasonably small width/mass ratio of 1%, such effects can be quite large, especially for large VLQ masses. The reason for these deviations, even with small widths, is primarily due to the presence of the aforementioned non-factorisable interference terms arising from topologies where the VLQ propagates non-resonantly in $t$-channel diagrams. These interference terms are usually neglected when applying the NWA, but they can in principle be relevant when the new physics couplings are large enough, and cannot be ignored [64, 65]. Their relative importance with respect to the NWA cross-section increases for increasing VLQ mass because the NWA cross-section drops much faster due to the need to produce $T$ resonantly.

For the case of single production the importance of interference effects for small width can be also seen in Fig.7 of [51] (and analogous for other processes in the same reference), where the isolines of constant $\hat{\sigma}$ become independent on the $\Gamma/M$ ratio when the width becomes smaller than certain values. We have checked that for smaller values of the width/mass ratio the relative differences with the NWA indeed fluctuate around zero or deviate at most of $\mathcal{O}(\%)$. This is consistent with the fact that the contribution to the cross-section associated to such interference diagrams scales quadratically with the new couplings, thus becoming less and less important as the couplings decrease.

To summarise, the results in Fig. 2 show that significant deviations from the SM background can be achieved both in pair and single production, allowing either production modes to be effective depending on the $T$ mass and width. For example, for a singlet $T$ with mass around 1.5 TeV (approximately the current limit for VLQs decaying to SM) and width around 1% of the mass, pair production processes with only top quarks in the final state are more likely to show significant deviation (more than 10%) from the SM background. If the width increases to 10% of the mass, single production with the Higgs boson in the final state feature similar or even larger deviations. For a doublet $T$ analogous conclusions hold. As the width increases, the relative importance of single production grows rapidly.

---

[4]All simulations in this and following sections have been done for LHC@13.6 TeV and at leading order using MG5_AMC [62] with the LO NNPDF 3.1 set of parton distribution functions [63]. The complex mass scheme has been adopted to treat the VLQ finite widths (more details about scheme dependence can be found in [51]). To allow for consistent comparisons, the renormalisation and factorisation scales have been set to the event-dependent sum of the transverse masses of final state particles divided by 2. Mild kinematic cuts have been applied ($p_{Tj,b} > 10$ GeV, $|\eta_{j,b}| < 5$ and $dR_{jj,jb,bb} > 0.01$).

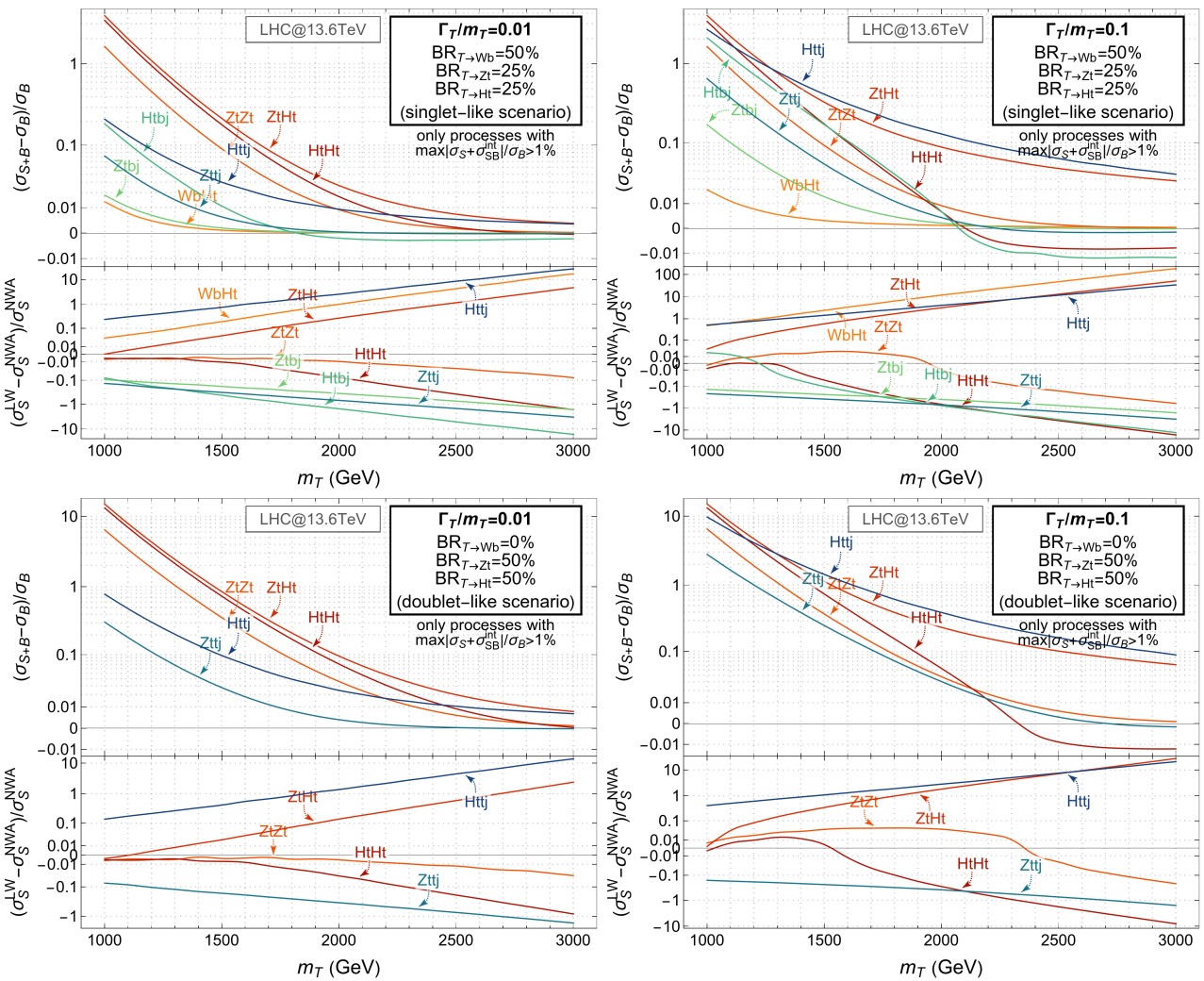

**Figure 2:** The four plots in this figure concern the singlet-like (**top row**) and doublet-like (**bottom row**) scenarios in the two cases where the VLQ width/mass ratios are 1% (**left column**) and 10% (**right column**) respectively. Each plot is split into an *upper panel* – showing the relative difference between the 13.6 TeV cross-sections of signal and background for each final state, and a *lower panel* – showing the relative difference between the signal cross-sections with the large width (LW) and those with the NWA. In the upper panels, the contribution of interference with background is included, leading to negative values for some processes in some $m_T$ ranges. In the lower panels, only processes for which the absolute value of the relative difference become larger than 1% in the range 1 TeV $\leq m_T \leq$ 3 TeV are shown.

The possibility to observe such deviations crucially relies on the design of signal regions capable of enhancing these deviations. The use of strong cuts on global variables can also help access higher mass scales where all channels contribute to generate signal events. These results also suggest the upper limit on systematic uncertainties that would ideally be needed to distinguish each signal from its irreducible background in four-particle final states.[5]

# 3   Exotic decays of VLQ

The main focus of this study is to explore the prospect for detecting various non-minimal scenarios involving VLQs at the LHC. As mentioned in the Introduction, the most notable form of non-minimality stems from the exotic decays of VLQs into BSM scalars and a third-generation quark. Specifically, VLQs can exhibit significant branching ratios to BSM scalars that emerge from various composite Higgs cosets, as illustrated in Tab. 1.

To make a quantitative statement regarding the pertinence of the exotic decay modes of the VLQs we illustrate it in a simple non-minimal extension of SM with a $SU(2)$ singlet vector-like top-partner ($T$) and a SM gauge singlet scalar ($S^0$). The next-to-minimal $SU(4)/Sp(4)$ coset space provides a concrete model where the above particle content can emerge. In Sec. 2, we already discussed the production modes of a singlet $T$, as well

---

[5]Including top or $W, Z, H$ boson decays and evaluating backgrounds for larger final state multiplicities goes beyond this simple estimate, and in that case a comprehensive investigation of backgrounds and reconstruction efficiencies would be essential for more precise analyses.

as its branching ratios in various SM final states in the absence of any BSM scalars. The relevant interaction Lagrangian to study the decays of $T$ is given by

$$\mathcal{L}_T = \frac{e}{\sqrt{2}s_W}\kappa_W^L \bar{T}\gamma^\mu W_\mu^+ P_L b + \frac{e}{2c_W s_W}\kappa_Z^L \bar{T}\gamma^\mu Z_\mu P_L t + \kappa_H^L H\bar{T}P_L t + \kappa_S^L S^0 \bar{T}P_L t + (L \leftrightarrow R) + \text{h.c.} \qquad (5)$$

The coupling strengths $\kappa^{L,R}$ can be parametrised up to $\mathcal{O}(v/m_T)$ as

$$|\kappa_W^L| = |\kappa_Z^L| = \frac{v}{m_T}\kappa, \quad |\kappa_W^R| = |\kappa_Z^R| = 0, \quad |\kappa_H^L| = \frac{m_T}{m_t}|\kappa_H^R| = \kappa, \quad |\kappa_S^L| = 0, \quad |\kappa_S^R| = \kappa_S, \qquad (6)$$

where $\kappa$ and $\kappa_S$ are two free parameters. In principle, (5) can be obtained from the Lagrangian (2) by identifying $\Psi \equiv T$ and $S \equiv S^0$, and diagonalizing the $T-t$ mass matrix. Note that, $\kappa_{W,Z}^L$ and $\kappa_H^R$ are suppressed by a factor of $\mathcal{O}(v/m_T)$ compared to $\kappa_S$. In addition, $\kappa_S^L$ is suppressed compared to $\kappa_S^R$ by $\mathcal{O}(v/m_T)$, and thus neglected in the following for simplicity.

Above the threshold $m_T > m_S + m_t$, the partial widths of $T$ into $S^0 t$ and SM final states are:

$$\Gamma_{T \to Wb} = \frac{\kappa^2 m_T}{16\pi}\lambda^{\frac{1}{2}}\left(1, \frac{m_b^2}{m_T^2}, \frac{M_W^2}{m_T^2}\right)\left[\left(1 - \frac{m_b^2}{m_T^2}\right)^2 + \frac{M_W^2}{m_T^2} - 2\frac{M_W^4}{m_T^4} + \frac{m_b^2 M_W^2}{m_T^4}\right], \qquad (7)$$

$$\Gamma_{T \to Zt} = \frac{\kappa^2 m_T}{32\pi}\lambda^{\frac{1}{2}}\left(1, \frac{m_t^2}{m_T^2}, \frac{M_Z^2}{m_T^2}\right)\left[\left(1 - \frac{m_t^2}{m_T^2}\right)^2 + \frac{M_Z^2}{m_T^2} - 2\frac{M_Z^4}{m_T^4} + \frac{m_t^2 M_Z^2}{m_T^4}\right], \qquad (8)$$

$$\Gamma_{T \to Ht} = \frac{\kappa^2 m_T}{32\pi}\lambda^{\frac{1}{2}}\left(1, \frac{m_t^2}{m_T^2}, \frac{m_H^2}{m_T^2}\right)\left[\left(1 + \frac{m_t^2}{m_T^2}\right)^2 + 4\frac{m_t^2}{m_T^2} - \frac{m_H^2}{m_T^2} - \frac{m_t^2 m_H^2}{m_T^4}\right], \qquad (9)$$

$$\Gamma_{T \to S^0 t} = \frac{\kappa_S^2 m_T}{32\pi}\lambda^{\frac{1}{2}}\left(1, \frac{m_t^2}{m_T^2}, \frac{m_S^2}{m_T^2}\right)\left[1 + \frac{m_t^2}{m_T^2} - \frac{m_S^2}{m_T^2}\right], \qquad (10)$$

where $\lambda^{1/2}(a,b,c) = \sqrt{a^2 + b^2 + c^2 - 2ab - 2ac - 2bc}$ denotes the Källén function. The relation between the branching ratio of the new decay channel $T \to S^0 t$ with respect to the SM decay channels $T \to Wb, Zt, Ht$ is:

$$BR_{T \to S^0 t} : BR_{T \to Wb} : BR_{T \to Zt} : BR_{T \to Ht} \simeq \frac{\kappa_S^2}{\kappa^2} : 2 : 1 : 1, \quad \text{for} \ \ m_S \ll m_T. \qquad (11)$$

Therefore, for $m_S \ll m_T$, the $T \to S^0 t$ decay can have a branching ratio as large as 50% if $\kappa_S \simeq 2\kappa$. For generic values of $m_S$, $BR_{T \to S^0 t}$ is determined by two ratios $\kappa_S/\kappa$ and $m_S/m_T$, as illustrated in Fig. 3.

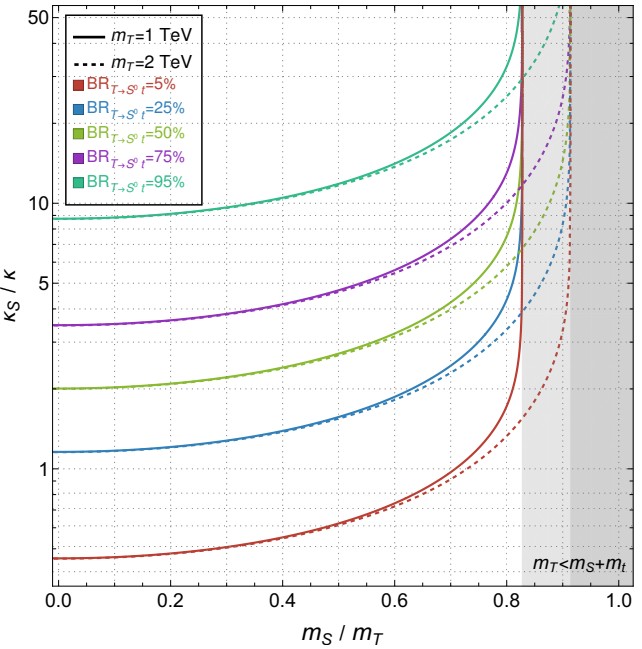

**Figure 3:** Fixed contours of the $BR_{T \to S^0 t}$ are shown in the $\kappa_S/\kappa - m_S/m_T$ plane for two different values of the VLQ mass, $m_T = 1$ TeV (solid lines) and $m_T = 2$ TeV (dashed lines).

A phenomenologically relevant question when VLQs have sizeable interactions with new scalars is which process(es) are most likely to produce signal events at the LHC. As described in Sec. 2, VLQs can be produced

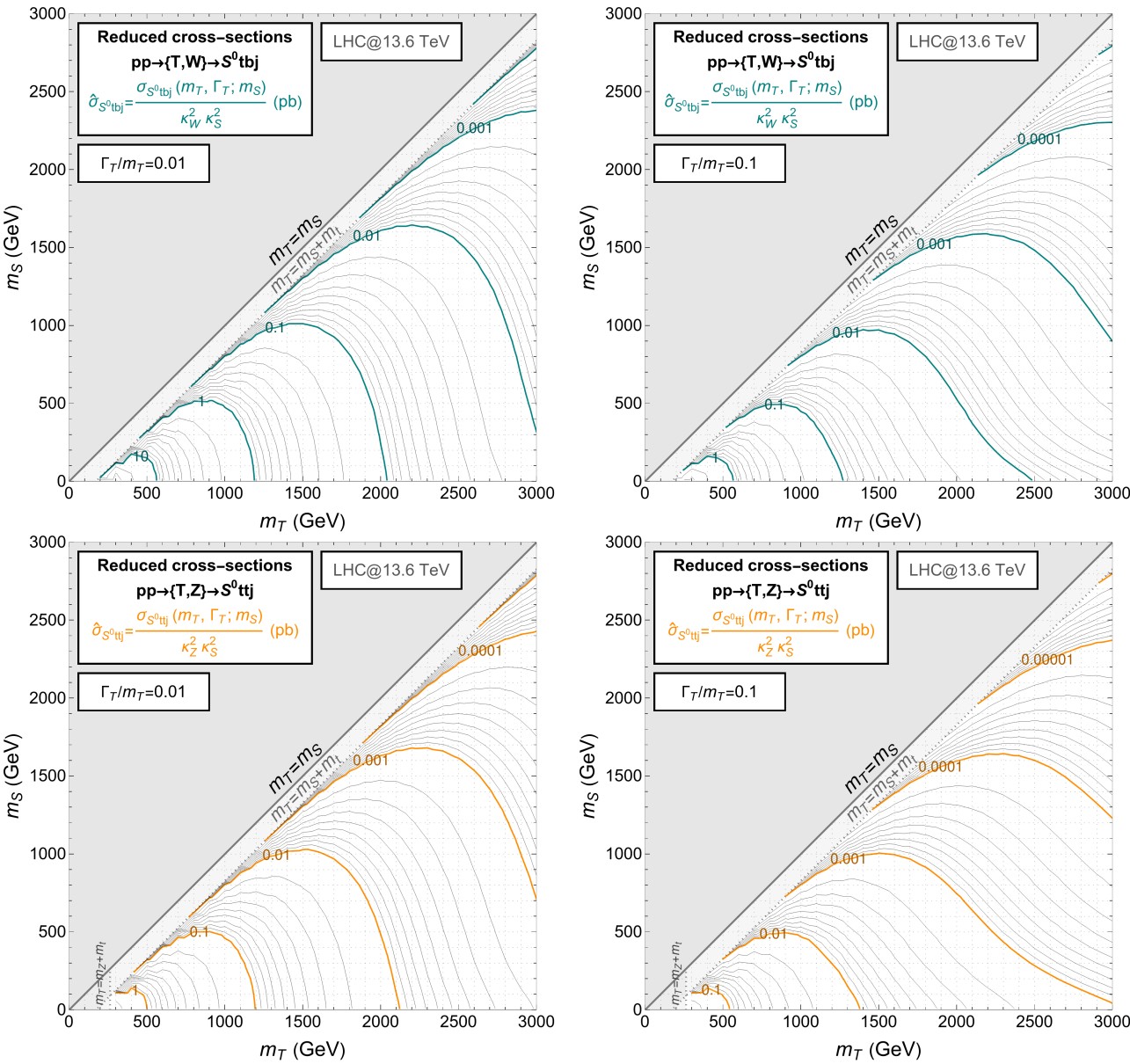

**Figure 4:** Reduced cross-sections $\hat{\sigma}$ for processes of single $T$ production via interactions with either $Wb$ (**top row**) or $Zt$ (**bottom row**) and in all cases decaying to a neutral scalar $S^0$ and the top quark. In the **left column** the $\Gamma_T/m_T$ ratio is 1%, while in the **right column** is fixed to 10%. Subsequent decays of the scalar have not been considered here.

in pairs or singly. The relative importance of these production processes crucially depends on the size of the interactions between VLQs and the SM quarks, and thus on the width of the VLQ. In case of exotic decays, the $\kappa_S$ coupling and the scalar mass $m_S$ can significantly affect the width. Thus, the summations in (3) and (4) should be augmented with the new decay channels.

Since pair production cross-sections mainly depend on the $T$ mass, and there are no interference contributions with the SM background in $2 \to 4$ processes involving new scalars in the final state, the cross-sections of these processes are simply scaled by the branching ratios to the exotic decay channels. Hence, we will focus on single production processes in the following discussion. We will assume that $T$ cannot be produced through its coupling with the new scalar $S^0$, for example because $S^0$ only interacts with heavy quarks or with SM bosons, and consider VLQ production mediated by the $W$ and $Z$ bosons only. The reduced cross-sections for the $S^0 t$ final state depend on the $T$ and $S^0$ masses, and on the total width of $T$, and are defined as:

$$\sigma_{S^0 tbj}(m_T, \Gamma_T; m_S) = (\kappa_W^L)^2 (\kappa_S^R)^2 \, \hat{\sigma}_{S^0 tbj}(m_T, \Gamma_T; m_S) \,, \tag{12a}$$

$$\sigma_{S^0 ttj}(m_T, \Gamma_T; m_S) = (\kappa_Z^L)^2 (\kappa_S^R)^2 \, \hat{\sigma}_{S^0 ttj}(m_T, \Gamma_T; m_S) \,. \tag{12b}$$

Their values values are shown in the $m_T - m_S$ plane in Fig. 4 for two choices of the $\Gamma_T/m_T$ ratio, namely 0.01 and 0.1.

We now have all the elements to assess which single production channels to consider in the presence of exotic

decays, and when signals from single production processes cease to be observable at all. Let us consider, as a working example, $m_T = 1.5$ TeV, $\Gamma_T/m_T = 1\%$ and $10\%$, $m_S = 500$ GeV, and the branching ratios of singlet-like $T$ varies as in (11). Using the couplings given in Eq. (6), reduced cross-sections for $S^0 t$ from Fig. 4 and those for the SM final states[6], we can derive the cross-sections associated with the individual single production processes. In Fig. 5 we show the comparison between such cross-sections as function of $BR_{T\to S^0 t}$. From these

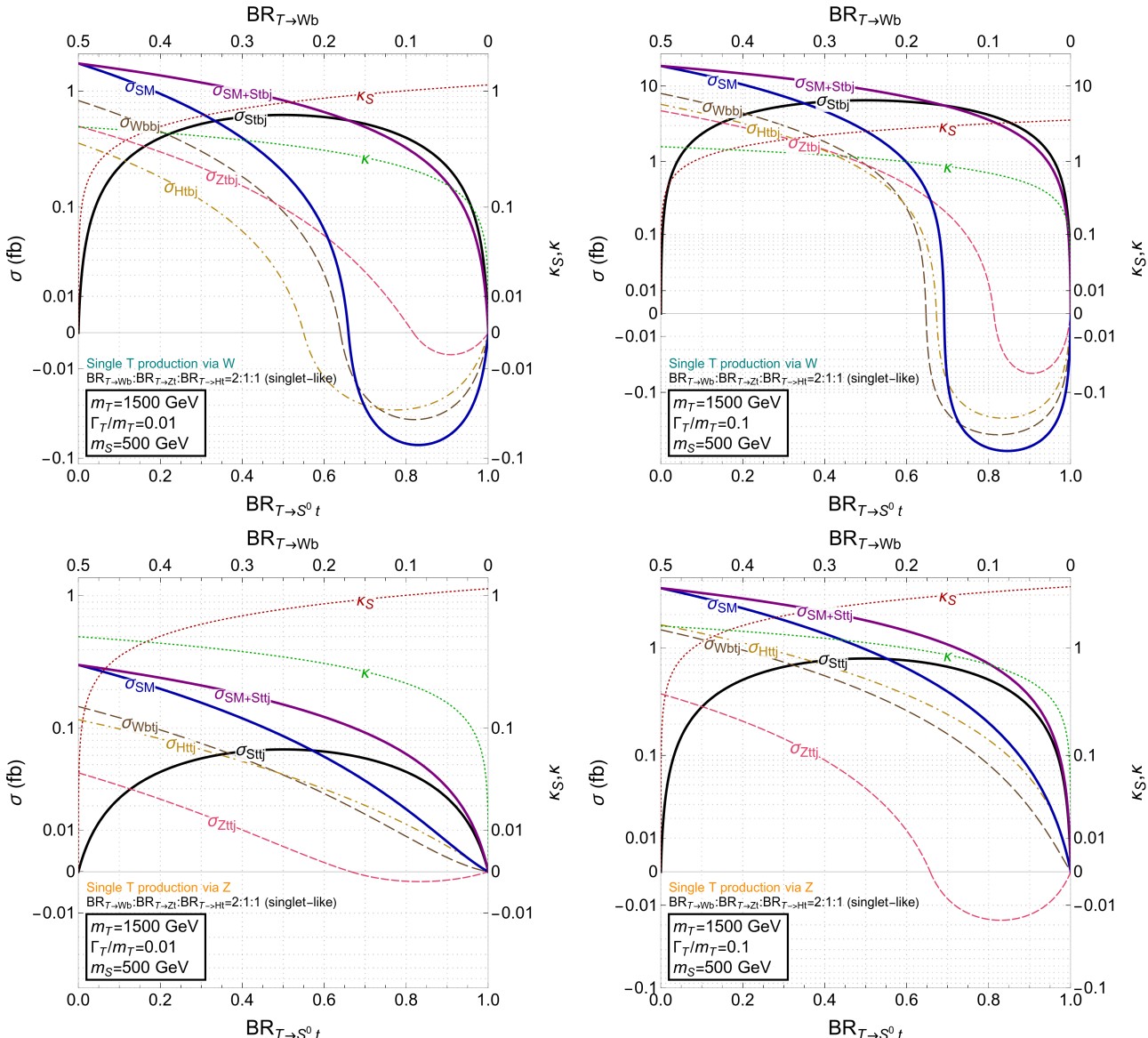

**Figure 5:** Single $T$ production cross-sections and their combined sums as function of $BR_{T\to S^0 t}$ for $m_T = 1500$ GeV and $m_S = 500$ GeV. For the SM final states, the contribution of interference with the SM background described in (12) has been included. The corresponding interaction couplings from (5) are also shown. In the **left column** $\Gamma_T/m_T = 1\%$, in the **right column** $\Gamma_T/m_T = 10\%$. Single production processes initiated by the interaction $TWb$ with coupling $\kappa_W^L$ are shown in the **top row**, while in the **bottom row** they are initiated by the interaction $TZt$ with coupling $\kappa_Z^L$. The negative cross-section values correspond to region where *negative interference* with the SM background has a dominant contribution.

results it is possible to infer a number of potentially relevant phenomenological consequences:

- The cross-section corresponding to single $T$ production going to the exotic $S^0 t$ final state is *always* maximal when $BR_{T\to S^0 t} = 0.5$, although its value also depends on the masses and widths configuration and if the process is initiated by interactions with $W$ or $Z$ bosons. In contrast, the cross-section becomes negligible in the opposite two limits: when $BR_{T\to S^0 t} = 0$, corresponding to $\kappa_S = 0$, and when $BR_{T\to S^0 t} = 1$, corresponding to $\kappa = 0$ (recall that we assume $T$ is singly produced solely through its interactions with the SM gauge bosons, not via $S^0$ itself, see Fig. 1).

---

[6]These have been obtained following the procedure in [51], but for LHC@13.6 TeV with the same cuts and PDF sets described in Sec. 2.

- The cross-sections to purely SM final states become subdominant when $BR_{T \to S^0 t}$ becomes larger than a specific value, which varies depending on the SM final state.

- Depending on the configurations of masses and widths, $\kappa_S$ may reach large values, making a perturbative approach to the calculation questionable.

- Some interference contributions for SM final states are negative,[7] and for $W$-initiated processes the SM combined sum $\sigma_{SM} = \sigma_{Wbbj} + \sigma_{Ztbj} + \sigma_{Htbj}$, represented by the blue curves in the plots, becomes negative when $BR_{T \to S^0 t}$ becomes larger than some value which depends on the masses and width. From Eq. (12), it is possible to see that interferences scale as $\kappa^2$, while the contributions from squaring signal topologies scale as $\kappa^4$. As $BR_{T \to S^0 t} \to 1$, $\kappa \to 0$, so that interference becomes eventually dominant and, when negative, it leads to a deficit of signal events in the SM channels.

- The sum of cross-sections for all SM and exotic final states, represented by the purple lines in the plots, is important when considering analyses based on particle multiplicities and global kinematic variables (such as $H_T$ or $H_T + \not{E}_T$), which are more agnostic to the specific decays of $S^0$. A significant contribution from the exotic final state to the sum is present even for relatively small values of $BR_{T \to S^0 t}$.

Apart from the above example other prominent exotic decay modes for VLQs include

$$T \to S^+ b, \quad B \to S^0 b, S^- t, \quad X_{5/3} \to S^+ t, S^{++} b, \quad Y_{-4/3} \to S^- b, S^{--} t, \quad \tilde{Y}_{8/3} \to S^{++} t. \tag{13}$$

In the next section, we will summarise the results from existing phenomenological analyses which impose limits on the masses of pair produced VLQs at the LHC, exploiting their BSM decay channels. Although the couplings with BSM scalars do not contribute to the single production of the VLQs at leading order, exotic decay modes may lead to novel search topologies for the singly produced VLQs. In Appendix B we present a comprehensive list of final states for a vector-like top partner ($T$) arising from both single and pair production processes, taking into account the two-body decays of $T$ into SM final states and new scalars which subsequently decay into SM particles.

Exotic decays can also play a crucial role in the direct search for VLQs with unusual electric charges, leading to distinctive smoking gun signatures. For instance, in the absence of any BSM scalars, $\tilde{Y}_{8/3}$ can only undergo a 3-body decay into $W^+ W^+ t$ via an off-shell $X_{5/3}$ exchange [30]. However, if a light doubly charged scalar is present, a novel 2-body decay channel, $\tilde{Y}_{8/3} \to S^{++} t$, with nearly 100% branching ratio can open up providing additional ways for detection [29].

We mention in passing, that the BSM scalars typically decay into dibosons or a pair of third generation quarks in the composite Higgs models, however, they can, in general, have additional decay channels into lighter quarks and leptons [40].

# 4 Limits from LHC Run 2 data

All published experimental limits on the masses of VLQs from the ATLAS and CMS collaborations assume that the VLQs undergo 2-body decays exclusively to SM particles. Limits on the masses of VLQs from searches for the pair- and single-production are summarised in Figs. 6 and 7, respectively.

Fig. 6 is an updated version of the plot presented in [26] (we kept only VLQ exclusion limits, updated experimental bounds and added other exotic channels) and shows a summary of several phenomenological analyses assuming exclusive decays (100% branching ratio) of VLQs into pNGBs and third generation quarks [18, 37, 40, 66]. When BSM decay channels are accessible, constraints on VLQ masses are considerably relaxed compared to current experimental searches. The weakest constraints are obtained in the case when the vector-like top decays to a pseudoscalar pNGB, which further decays to two photons or a photon and a $Z$ boson [18]. In contrast, the decay of $X_{5/3} \to S^{++} b$, provides the strongest limit when the $S^{++}$ undergoes a lepton number violating decay into $S^{++} \to e^+ e^+$ [40].

The strongest limits obtained from published experimental searches for pair-produced VLQs decaying into SM particles are shown as horizontal bars. These limits assume 100% branching ratios to the corresponding decay mode. The most stringent lower limits on the VLQ mass were found to be 1.7 TeV in a vector-like top search in the $T \to Wb$ decay mode [70] and 1.56 TeV in a vector-like bottom search in the $B \to Hb$ mode [67], using the data collected during the full Run 2 of the LHC. Since $Y_{-4/3}$ decays only to $Wb$ assuming purely SM decays, the strongest bound on $T$ reported in [70] also applies to $Y_{-4/3}$ since this limit assumes $BR_{T \to Wb} = 100\%$[8].

---

[7] We stress that the sum of signal and interference *can* be negative, as long as the total cross-section, defined as the sum of contributions from SM background, signal and their interference, is positive.

[8] For pair production it is possible to reinterpret limits on $T$ for $Y_{-4/3}$ even if $BR_{T \to Wb} < 100\%$, assuming there's no contamination from other channels. For single production, if $BR_{T \to Wb} < 100\%$, such a reinterpretation is not trivial because the production cross-section also depends on the same couplings which determine the BRs.

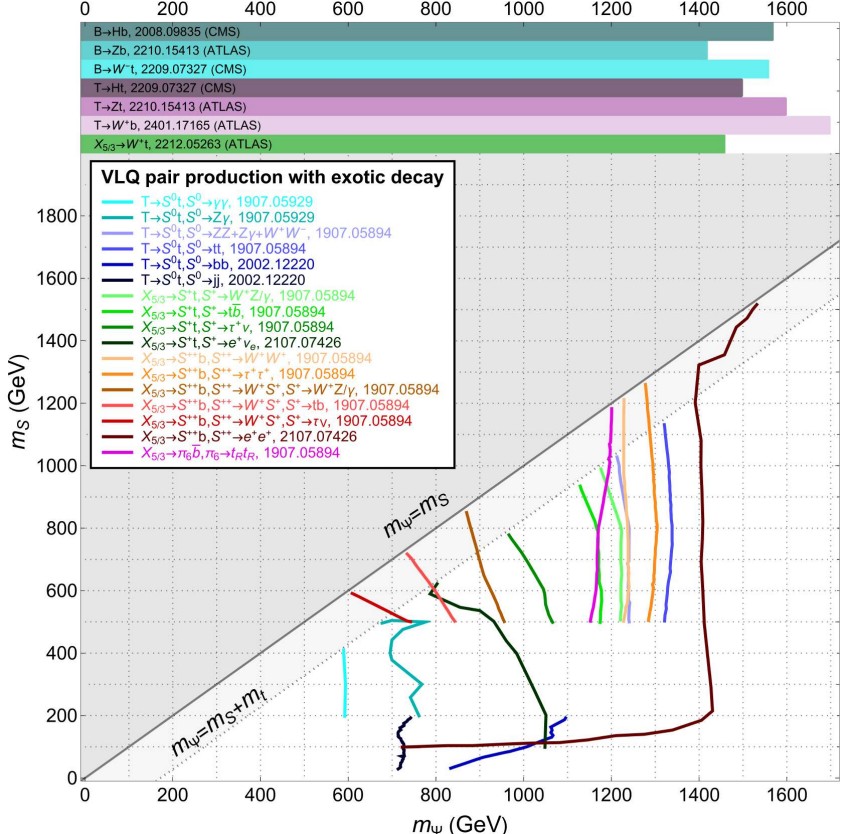

**Figure 6:** 95% CL exclusion limits on the $m_\Psi - m_S$ plane from pair production of VLQs. The horizontal bars at the top show current experimental limits from ATLAS and CMS for VLQ decays into SM final states with 100% branching ratio. The solid lines show recast limits on different exotic decays using LHC Run-2 data. The area to the left of each curve is excluded by the current data. This figure is an updated version of the summary plot presented in [26]. References used for VLQ pair production with exotic decay: [18, 37, 40, 66], and for VLQ limits from SM decays: [67–71].

Mass limits on singly produced VLQs depend on their coupling to the $W$ and $Z$ bosons, their mass, and their width. Different searches employ different ways to report the observed limits:

1. limits on the production cross-section as a function of $m_\Psi$ (e.g. [72])

2. limits on the couplings of $\Psi$ to the $W$, $Z$ and Higgs bosons as a function of $m_\Psi$ (e.g. [52])

3. limits on the ratio of width over mass $\Gamma_\Psi / m_\Psi$ as a function of $m_\Psi$ (e.g. [56])

Limits obtained from various ATLAS and CMS searches for singly produced VLQs using 139 fb$^{-1}$ of data collected during Run 2 of the LHC are summarised in Fig. 7. The $\Gamma_\Psi / m_\Psi$ vs $m_\Psi$ plane is used to make this comparison since the event kinematics depend only on the total width and the mass of the VLQ. The mapping between the EW couplings and $\Gamma_\Psi / m_\Psi$ is feasible and has been used when the limits are not available in the $\Gamma_\Psi / m_\Psi$ vs $m_\Psi$ plane. However, to map the bounds on the cross-sections to this plane, simulations for a grid in $m_\Psi$ and $\Gamma_\Psi$ needs to be performed for each VLQ representation, which is resource intensive and beyond the scope of this work.

As shown in Fig. 7, the lower bounds on $m_\Psi$ range from $700 - 2000$ GeV depending on the width of the VLQ. The strongest bounds at masses above 1.5 TeV are obtained in an ATLAS search targeting $T \to Ht, Zt$ in the one-lepton final state. Both ATLAS and CMS have performed a search targeting the $T \to Zt, Z \to \nu\nu$ decay mode in the 0-lepton final state. However, the bounds reported by ATLAS are stronger than those reported by CMS by about 500 GeV. While the basic selection, (such as cuts on the $p_T$ of large-R jets, and the total hadronic energy), are similar in both searches, the search performed by ATLAS employs an extreme gradient Boosted Decision Trees trained on VLQ masses above 1.5 TeV, resulting in significantly stronger bounds.

In general, the bounds are stronger on VLQs with larger widths since the cross-section for single production of VLQs increases with width. While the summary is not exhaustive, it faithfully represents the trend in every search. The two most crucial observations are:

- The bounds on vector-like top in the doublet representation (with branching ratio pattern shown in Sec. 2) are significantly weaker than the singlet case given the forbidden coupling of the VLQ with the $W$ bosons, which results in a lower production cross-section.

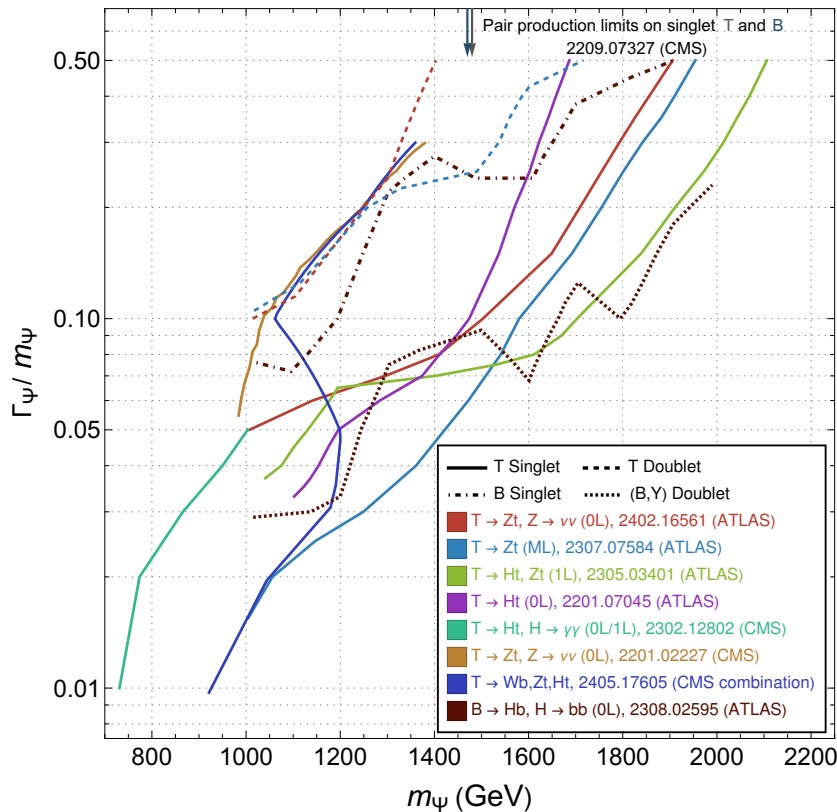

**Figure 7:** 95% CL exclusion limits from the ATLAS and CMS full Run 2 data on the $\Gamma_\Psi/m_\Psi - m_\Psi$ plane from single production of VLQs and their decays into SM final states [50, 52–57]. The area to the left of each curve is excluded by the current data. For comparison, the strongest lower bound on the mass of pair-produced singlet/doublet VLQ is also indicated. The difference in the bounds on the mass of pair-produced VLQs between the singlet and doublet cases is only about 10 GeV and indistinguishable in this figure.

- The LHC data is not sensitive to singly-produced narrow-width vector-like tops with a high mass. Thus, limits from searches for singly-produced VLQs significantly improve on the pair-production searches only in the large width regime.

Only the strongest experimental bounds are discussed in this section. In Appendix A, we list all relevant searches for VLQs performed by the ATLAS and CMS collaborations at Run 2.

# 5    Looking ahead: LHC Run 3 and HL-LHC

So far, VLQ searches at the LHC have focused on final states resulting from their 2-body decays into SM particles, either through pair or single production modes. However, several well-motivated BSM scenarios, such as CHMs and models addressing flavour observables, predict additional BSM scalars that lead to exotic VLQ decays and increase their total widths.

In Secs. 2 and 3, we have presented quantitative analyses using a VLQ $T$ as an example to address the issues arising in VLQ production in the large width regime or with non-SM decays, respectively. In particular, a systematic computation of the pair production cross-section, accounting for significant interference effects, is essential for large widths. Additionally, we have illustrated the relevance of exotic decay channels compared to the SM ones and summarise current limits on VLQ masses from their exotic decays based on existing phenomenological analyses. We employ a signal parametrisation that is suitable to describe a wide range of signals and reinterpret experimental results in any new physics scenario predicting VLQs.

To summarise the results from the present experimental searches, as shown in Sec. 4, the limit on the VLQ masses is around 1.5 TeV, the exact limit being dependent on the particulars of searches, targeted final states and the widths of the VLQs. The general trend indicates that limits from the single production becomes competitive with the pair production only at large widths.

In the future, pair production searches will be constrained by energy at high VLQ mass, while single production will be limited by statistics for moderate to low widths. Insights from Run 2 searches will be crucial in shaping future strategies. Additionally, novel topologies and observables should be explored to distinguish

signals from the background despite limited phase space and statistics. We now highlight some additional aspects to consider for future searches at Run 3 and HL-LHC.

**Multiple VLQs:** Minimal extensions of the SM are appealing due to their simplicity and limited new parameters, making them ideal for new physics studies. However, most new physics scenarios, such as those mentioned in Sec. 1, predict multiple VLQs belonging to different representations containing VLQs with different charges (including exotic ones), but with nearly degenerate mass. The presence of multiple VLQs can help evading precision and flavour bounds even with large couplings (and thus large widths) [60, 61]. The presence of more than one VLQ too heavy to be observed individually would provide more signal events when combined together to populate experimental signal regions, pushing their sensitivity reach to higher VLQ masses [29]. In this case, the interplay of single and pair production would provide complementary information which can be exploited to find possible excesses.

**New production channels:** Apart from the conventional production modes, considered in Sec. 2, new production channels for the VLQs may be present in non-minimal models (for e.g. [73–75]). Some existing experimental searches, primarily conducted by the CMS collaboration [76–80], have leveraged the decays of additional BSM particles, such as $W'$ or $Z'$, to produce a single VLQ in association with a third generation quark. Pair production of VLQs may also receive contributions from the decays of colour octet gluon partners [81–83]. Further, higher dimensional operators such as the chromomagnetic operator (see Tab. 2), suppressed by the new physics scale, may provide extra contributions to the single VLQ production [84, 85].

**Exotic decays:** Interactions of VLQs with comparatively lighter BSM bosons (scalars or vectors of various charges) may contribute to their total widths through exotic decay channels, as extensively discussed in Sec. 3. This expands the range of possible final states when producing VLQs. Such final states might also populate signal regions designed for VLQs decaying into SM particles, though their unique kinematic properties could necessitate the definition of new signal regions with better sensitivity. A crucial aspect of exotic interactions is that interactions with SM particles might be subdominant [28]. This in turn implies that single production, driven by SM couplings, may not be the optimal channel for exploring VLQs with exotic interactions, making pair production important, even at high mass. It is important to highlight that a `FeynRules` model file, capable of incorporating multiple VLQs and BSM scalars, and the exotic VLQ decays at the NLO QCD precision, is available publicly [26][9].

**Interactions with light quarks:** While composite-Higgs motivated scenarios of new physics privilege interactions of VLQs with top and bottom quarks, from a model-agnostic point of view it would be interesting to consider the possibility of interactions with lighter generations [7, 86–88]. Such scenarios can also be theoretically motivated [89, 90]. These have been explored experimentally in QCD pair production by ATLAS at 7 and 8 TeV [91, 92] and very recently with full 13 TeV Run 2 data [93]. While being more heavily constrained by flavour observables and electroweak precision measurements, they can lead to peculiar signatures in both single and pair production: for example, a new signature such as same-sign VLQ production [60, 94] has not been explored experimentally, either with SM or exotic decays.

**Experimental strategies:** While the presence of multiple VLQs can increase the production cross-section of the signal, their exotic decays result in diverse final states with varying multiplicities of photons, and $W$, $Z$ bosons arising from the bosonic decays of the BSM scalar. Including the fermionic decays of the $S$, the final states can consist of 0 to 6 leptons (described in Appendix B). Traditionally, searches for SM decays of VLQs have focused on $0\ell$ or $1\ell$ final states, with specific requirements on kinematic variables such as $p_T$ of jets, and global variables such as the effective mass. Since the limits on $m_\Psi$ are rather high, the experimental searches have also required the presence of large-radius jets originating from top quarks, the Higgs boson, and $W$ and $Z$ bosons. These strategies to search for VLQs have applicability while considering new possibilities such as those mentioned above. Global variables such as the effective mass have been shown to be efficient in discriminating between signal and background also in cases when a VLQ decays to BSM scalars [18, 29]. Boosted objects, resulting in large-radius jets, may arise if the difference $\Delta m \equiv (m_\Psi - m_S)$ is substantial. In such cases, the algorithms to identify large-radius jets can be further optimised for the phase space corresponding to large $\Delta m$. Furthermore, given the increased production cross-section from multiple VLQs, it would be important to consider final states with same-sign dileptons, at least three leptons, and photons, as these have low SM backgrounds and can enhance discovery potential.

---

[9]Vector-like quarks + exotic pNGBs: https://feynrules.irmp.ucl.ac.be/wiki/NLOModels

To conclude, searches of VLQs can still follow multiple paths, both considering pair and single production. Numerous signatures are yet to be explored, which lead to reconsidering current bounds, and there is potential to push the reach of sensitivity to much higher masses.

# Acknowledgements

This work would not have been possible without financing from the Knut and Alice Wallenberg foundation under the grant KAW 2017.0100, during the years 2017-2023. We would like to thank D. B. Franzosi, G. Cacciapaglia, A. Deandrea, T. Flacke, B. Fuks, M. Kunkel, W. Porod, and L. Schwarze for previous collaborations on related matters. A.B. and G.F. would like to express special thanks to the Mainz Institute for Theoretical Physics (MITP) of the Cluster of Excellence PRISMA+ (Project ID 390831469), for its hospitality and support. A.B. would like to thank the Chalmers University of Technology, Göteborg, Sweden for support during the initial stages of this work, and acknowledges support from the Department of Atomic Energy, Govt. of India. G.F. is partly supported by grants from the Wilhelm och Martina Lundgren foundation, and the Adlerbert Research Foundation via the KVVS foundation. L.P.'s work is supported by ICSC – Centro Nazionale di Ricerca in High Performance Computing, Big Data and Quantum Computing, funded by European Union – NextGenerationEU.

# A    List of ATLAS and CMS VLQ papers

## A.1    Pair-production

All searches for pair-produced VLQs at the Run 2 of the LHC are summarised in Tab. 4, excluding results with less than $5\,\mathrm{fb}^{-1}$ since significantly stronger results are obtained with more data. In addition, the ATLAS collaboration has reported results by combining several studies with $36\,\mathrm{fb}^{-1}$ of data [95], which show that singlet $T(B)$ below 1.31(1.22) TeV and $(T, B)$ doublets below 1.37 TeV are excluded. Many searches target $0\ell$ and $1\ell$ final states since signal yield drops significantly with stronger requirements on the number of leptons. However, searches including same-sign dilepton ($2\ell SS$) and three-lepton ($3\ell$) final states can enhance the sensitivity of VLQ searches (for example, see [69]), especially in models with multiple VLQs, where higher production cross-sections compensate for the loss of signal from suppressed branching ratios of leptonic decays of $W/Z$ bosons.

| VLQ | Decay | Experiment | Dataset | Lepton multiplicity | Singlet Limit (in TeV) | Doublet Limit (in TeV) |
|-----|-------|-----------|---------|--------------------|------------------------|------------------------|
| $T, B$ | Inclusive | ATLAS [70] | $140\,\mathrm{fb}^{-1}$ | $1\ell$ | 1.36, 1.21 | $-, 1.345$ |
| | Inclusive | ATLAS [71] | $139\,\mathrm{fb}^{-1}$ | $1\ell$ | 1.26, 1.33 | 1.59, 1.59 |
| | $Z(\ell\ell)t, Z(\ell\ell)b$ | ATLAS [68] | $139\,\mathrm{fb}^{-1}$ | $2\ell OS, 3\ell$ | 1.27, 1.20 | 1.46, 1.32 |
| | Inclusive | CMS [69] | $138\,\mathrm{fb}^{-1}$ | $1\ell, 2\ell SS, 3\ell$ | 1.48, 1.47 | 1.49, 1.12 |
| | Inclusive | ATLAS [96] | $36.1\,\mathrm{fb}^{-1}$ | $0\ell$ | $-$ | $1.01*, 0.95$ |
| | Inclusive | CMS [97] | $36.1\,\mathrm{fb}^{-1}$ | $0\ell$ | $1.37*, -$ | $-, 1.23$ |
| | $Wb, Wt$ | ATLAS [98] | $36.1\,\mathrm{fb}^{-1}$ | $0\ell$ | 1.17, 1.08 | 0.75, 1.25 |
| | $Z(\ell\ell)t, Z(\ell\ell)b$ | ATLAS [99] | $36.1\,\mathrm{fb}^{-1}$ | $2\ell OS, \geq 3\ell$ | 1.03, 1.01 | 1.21, 1.14 |
| | Inclusive | ATLAS [100] | $36.1\,\mathrm{fb}^{-1}$ | $2\ell SS$ | 0.98, 1.0 | $-$ |
| | Inclusive | CMS [101] | $35.9\,\mathrm{fb}^{-1}$ | $1\ell, 2\ell SS, 3\ell$ | 1.2, 1.17 | 1.28, 0.94 |
| | $Z(\ell\ell)t, Z(\ell\ell)b$ | CMS [102] | $35.9\,\mathrm{fb}^{-1}$ | $2\ell OS$ | $-$ | $1.28*, 1.13*$ |
| $T$ | $Z(\nu\nu)t$ | ATLAS [103] | $36.1\,\mathrm{fb}^{-1}$ | $1\ell$ | 0.87 | 1.05 |
| | $Ht, Zt$ | ATLAS [104] | $36.1\,\mathrm{fb}^{-1}$ | $1\ell$ | 1.19 | 1.31 |
| $B$ | $Hb, Zb$ | CMS [67] | $138\,\mathrm{fb}^{-1}$ | $0\ell$ | 1.1 | $-$ |
| | Inclusive | CMS [105] | $138\,\mathrm{fb}^{-1}$ | $0\ell, 2\ell SS$ | 1.06 | 1.5 |
| | $Wt$ | ATLAS [106] | $36.1\,\mathrm{fb}^{-1}$ | $1\ell$ | 1.17 | 1.35 |
| $X$ | $Wt$ | ATLAS [71] | $139\,\mathrm{fb}^{-1}$ | $1\ell$ | 1.59 | |
| | | ATLAS [106] | $36.1\,\mathrm{fb}^{-1}$ | $1\ell$ | 1.35 | |
| | | ATLAS [98] | $36.1\,\mathrm{fb}^{-1}$ | $0\ell$ | 1.25 | |
| | | ATLAS [100] | $36.1\,\mathrm{fb}^{-1}$ | $2\ell SS$ | 1.19 | |
| | | CMS [107] | $35.9\,\mathrm{fb}^{-1}$ | $1\ell, 2\ell SS$ | $1.33*$ | |
| $Y$ | $Wb$ | ATLAS [70] | $140\,\mathrm{fb}^{-1}$ | $1\ell$ | 1.7 | |
| | | ATLAS [98] | $36.1\,\mathrm{fb}^{-1}$ | $0\ell$ | 1.35 | |
| | | CMS [108] | $35.8\,\mathrm{fb}^{-1}$ | $1\ell$ | 1.295 | |
| $Q$ | $Wq$ | ATLAS [93] | $140\,\mathrm{fb}^{-1}$ | $1\ell$ | 1.15 | $1.53*$ |

**Table 4:** Summary of ATLAS and CMS pair-production VLQ searches for $T, B, X_{5/3}, Y_{-4/3}$. For di-lepton final states, a further distinction is made to differentiate between opposite-sign and same-sign leptons, denoted by $2\ell OS$ and $2\ell SS$ respectively. The type of doublet (for example $(T, B), (B, Y)$ etc) is not specified here, and can be different across the results summarised here. In cases when the published result does not report the singlet and doublet limits, the strongest limit is reported and marked with '*'. When the published results provide limits on more than one VLQ type, the limits are separated by a comma and reported in the same order as that of the VLQs in the first column. The $QQ \to WqWq$ search [93] refers to a VLQ coupling to lighter quarks ($q = u, d, s, c$). Limits obtained with data less than $5\,\mathrm{fb}^{-1}$ are not included, since all such studies have been superseded with other searches using more data for the same final state.

## A.2 Single production

All searches for singly-produced VLQs at $\sqrt{s} = 13$ TeV are summarised in Tab. 5. The limits obtained from searches which report the results in either the $\Gamma_\Psi/m_\Psi$ vs $m_\Psi$ plane or the EW coupling of $\Psi$ vs $m_\Psi$ plane are summarised in Fig. 7 and hence, the limits are not shown in Tab. 5. Searches for singly-produced VLQs using the entire dataset collected during Run 2 of the LHC do not cover all SM decays of the VLQ yet. Notably, the existing $T \to Wb$ searches only use partial Run 2 data. Recent summary papers from ATLAS and CMS give comprehensive overviews of the status of single VLQ searches in the experiments [49, 50]. In addition, the CMS collaboration has also reported limits on $m_\Psi$ in BSM production modes, $pp \to W'/Z' \to B/T + t$ [76–79] and on excited quarks [109–112].

| VLQ | Decay | Experiment | Dataset | Lepton multiplicity |
|---|---|---|---|---|
| $T$ | $Z(\nu\nu)t$ | ATLAS [55] | $140\,\mathrm{fb}^{-1}$ | $0\ell$ |
| | $Z(\nu\nu)t$ | CMS [56] | $137\,\mathrm{fb}^{-1}$ | $0\ell$ |
| | $Z(\nu\nu)t$ | ATLAS [113] | $36.1\,\mathrm{fb}^{-1}$ | $0\ell, 1\ell$ |
| | $H(bb)t$ | ATLAS [52] | $139\,\mathrm{fb}^{-1}$ | $0\ell$ |
| | $H(\gamma\gamma)t$ | CMS [57] | $138\,\mathrm{fb}^{-1}$ | $0\ell, 1\ell$ |
| | $Ht, Zt$ | CMS [114] | $138\,\mathrm{fb}^{-1}$ | $0\ell$ |
| | $Ht, Zt$ | ATLAS [54] | $138\,\mathrm{fb}^{-1}$ | $1\ell$ |
| | $Ht, Zt$ | CMS [72] | $35.9\,\mathrm{fb}^{-1}$ | $0\ell$ |
| | $Z(\ell\ell)t$ | ATLAS [53] | $139\,\mathrm{fb}^{-1}$ | $2\ell OS, 3\ell$ |
| | $Z(\ell\ell)t$ | ATLAS [99] | $36.1\,\mathrm{fb}^{-1}$ | $2\ell OS, \geq 3\ell$ |
| | $Z(\ell\ell)t$ | CMS [80] | $35.9\,\mathrm{fb}^{-1}$ | $2\ell OS$ |
| $T, Y$ | $W(\ell\nu)b$ | ATLAS [115] | $36.1\,\mathrm{fb}^{-1}$ | $1\ell$ |
| | $W(\ell\nu)b$ | CMS [116] | $2.3\,\mathrm{fb}^{-1}$ | $1\ell$ |
| $B$ | $H(bb)b$ | ATLAS [117] | $139\,\mathrm{fb}^{-1}$ | $0\ell$ |
| | $H(bb)b$ | CMS [118] | $35.9\,\mathrm{fb}^{-1}$ | $0\ell$ |
| | $Z(\ell\ell)b$ | ATLAS [99] | $36.1\,\mathrm{fb}^{-1}$ | $2\ell OS, \geq 3\ell$ |
| $B, X$ | $Wt$ | CMS [119] | $35.9\,\mathrm{fb}^{-1}$ | $1\ell$ |

**Table 5:** Summary of ATLAS and CMS single VLQ searches for $T, B, X_{5/3}, Y_{-4/3}$.

# B List of final states for vector-like top search

A comprehensive list of final states arising from the single and pair production of a vector-like top $T$, followed by its standard and exotic decays are displayed in Tabs. 6 and 7. The final states are categorised on the basis of number of b-jets ($N_b$), total number of leptons ($N_l$), number of same sign lepton pairs ($N_{\mathrm{SSL}}$), and of photons ($N_\gamma$). We also present the number of intermediate $W^\pm, Z$, and $H$ bosons. The $W, Z, H$ bosons are assumed to further decay into $W^+ \to 2j, l^+\nu$, $Z \to 2j, b\bar{b}, l^+l^-, \not{E}_T$, and $H \to b\bar{b}$, respectively. The categorisation of final states arising from other VLQs can be done in similar way.

| Production | Contributing decay channels | | | Final state multiplicities | | | | | | |
|---|---|---|---|---|---|---|---|---|---|---|
| | | | | $N_{(W^+,W^-)}$ | $N_Z$ | $N_\gamma$ | $N_H$ | $N_b$ | $N_l$ | $N_{\mathrm{SSL}}$ |
| $T\bar{t}j$ | SM | $T \to W^+b$ | | $(1,1)$ | – | – | – | 2 | $\leq 2$ | – |
| | | $T \to Zt$ | | $(1,1)$ | 1 | – | – | $\leq 4$ | $\leq 4$ | $\leq 2$ |
| | | $T \to Ht$ | | $(1,1)$ | – | – | 1 | $\leq 4$ | $\leq 2$ | – |
| | BSM | $T \to S^0 t$ | $S^0 \to t\bar{t}$ | $(2,2)$ | – | – | – | 4 | $\leq 4$ | $\leq 2$ |
| | | | $S^0 \to b\bar{b}$ | $(1,1)$ | – | – | – | 4 | $\leq 2$ | – |
| | | | $S^0 \to W^+W^-$ | $(2,2)$ | – | – | – | 2 | $\leq 4$ | $\leq 2$ |
| | | | $S^0 \to ZZ$ | $(1,1)$ | 2 | – | – | $\leq 6$ | $\leq 6$ | $\leq 3$ |
| | | | $S^0 \to \gamma Z$ | $(1,1)$ | 1 | 1 | – | $\leq 4$ | $\leq 4$ | $\leq 2$ |
| | | | $S^0 \to \gamma\gamma$ | $(1,1)$ | – | 2 | – | 2 | $\leq 2$ | – |
| | | $T \to S^+ b$ | $S^+ \to t\bar{b}$ | $(1,1)$ | – | – | – | 3 | $\leq 2$ | – |
| | | | $S^+ \to W^+\gamma$ | $(1,1)$ | – | 1 | – | 2 | $\leq 2$ | – |
| | | | $S^+ \to W^+Z$ | $(1,1)$ | 1 | – | – | $\leq 4$ | $\leq 4$ | $\leq 2$ |
| $T\bar{b}j$ | SM | $T \to W^+b$ | | $(1,0)$ | – | – | – | 2 | $\leq 1$ | – |
| | | $T \to Zt$ | | $(1,0)$ | 1 | – | – | $\leq 4$ | $\leq 3$ | $\leq 1$ |
| | | $T \to Ht$ | | $(1,0)$ | – | – | 1 | $\leq 4$ | $\leq 1$ | – |
| | BSM | $T \to S^0 t$ | $S^0 \to t\bar{t}$ | $(2,1)$ | – | – | – | 4 | $\leq 3$ | $\leq 1$ |
| | | | $S^0 \to b\bar{b}$ | $(1,0)$ | – | – | – | 4 | $\leq 1$ | – |
| | | | $S^0 \to W^+W^-$ | $(2,1)$ | – | – | – | 2 | $\leq 3$ | $\leq 1$ |
| | | | $S^0 \to ZZ$ | $(1,0)$ | 2 | – | – | $\leq 6$ | $\leq 5$ | $\leq 2$ |
| | | | $S^0 \to \gamma Z$ | $(1,0)$ | 1 | 1 | – | $\leq 4$ | $\leq 3$ | $\leq 1$ |
| | | | $S^0 \to \gamma\gamma$ | $(1,0)$ | – | 2 | – | 2 | $\leq 1$ | – |
| | | $T \to S^+ b$ | $S^+ \to t\bar{b}$ | $(1,0)$ | – | – | – | 3 | $\leq 1$ | – |
| | | | $S^+ \to W^+\gamma$ | $(1,0)$ | – | 1 | – | 2 | $\leq 1$ | – |
| | | | $S^+ \to W^+Z$ | $(1,0)$ | 1 | – | – | $\leq 4$ | $\leq 3$ | $\leq 1$ |

**Table 6:** Categorisation of final states for single production of a vector-like top quark.

| Contributing decay channels | | $N_{(W^+,W^-)}$ | $N_Z$ | $N_\gamma$ | $N_H$ | $N_b$ | $N_l$ | $N_{\mathrm{SSL}}$ |
|---|---|---|---|---|---|---|---|---|
| $T\bar{T} \to W^+W^-b\bar{b}$ | | $(1,1)$ | – | – | – | 2 | $\le 2$ | – |
| $T\bar{T} \to ZZt\bar{t}$ | | | 2 | – | – | $\le 6$ | $\le 6$ | $\le 3$ |
| $T\bar{T} \to ZHt\bar{t}$ | | | 1 | – | 1 | $\le 6$ | $\le 4$ | $\le 2$ |
| $T\bar{T} \to HHt\bar{t}$ | | | – | – | 2 | $\le 6$ | $\le 2$ | – |
| | $S^0 \to t\bar{t}, b\bar{b}$ | $(\le 3, \le 3)$ | – | – | – | 6 | $\le 6$ | $\le 3$ |
| $T\bar{T} \to 2S^0 t\bar{t}$ | $S^0 \to$ bosonic | $(1,1)$ | 4 | – | – | $\le 10$ | $\le 10$ | $\le 5$ |
| | | | 3 | 1 | – | $\le 8$ | $\le 8$ | $\le 4$ |
| | | | 2 | 2 | – | $\le 6$ | $\le 6$ | $\le 3$ |
| | | | 1 | 3 | – | $\le 4$ | $\le 4$ | $\le 2$ |
| | | | – | 4 | – | 2 | $\le 2$ | – |
| | | $(2,2)$ | 2 | – | – | $\le 6$ | $\le 8$ | $\le 4$ |
| | | | 1 | 1 | – | $\le 4$ | $\le 6$ | $\le 3$ |
| | | | – | 2 | – | 2 | $\le 4$ | $\le 2$ |
| | | $(3,3)$ | – | – | – | 2 | $\le 6$ | $\le 3$ |
| | $S^+ \to t\bar{b}$ | $(1,1)$ | – | – | – | 6 | $\le 2$ | – |
| $T\bar{T} \to S^+S^-b\bar{b}$ | $S^+ \to$ bosonic | $(1,1)$ | 2 | – | – | $\le 6$ | $\le 6$ | $\le 3$ |
| | | | 1 | 1 | – | $\le 4$ | $\le 4$ | $\le 2$ |
| | | | – | 2 | – | 2 | $\le 2$ | – |
| | $S^0 \to t\bar{t}, b\bar{b}$ | $(\le 2, \le 2)$ | – | – | $\le 1$ | $\le 6$ | $\le 4$ | $\le 2$ |
| | | | 1 | – | – | $\le 6$ | $\le 6$ | $\le 3$ |
| $T\bar{T} \to S^0 W^- t\bar{b}$ | $S^0 \to$ bosonic | $(1,1)$ | 3 | – | – | $\le 8$ | $\le 8$ | $\le 4$ |
| $T\bar{T} \to S^0 Z t\bar{t}$ | | | 2 | $\le 1$ | $\le 1$ | $\le 8$ | $\le 6$ | $\le 3$ |
| $T\bar{T} \to S^0 H t\bar{t}$ | | | 1 | $\le 2$ | $\le 1$ | $\le 6$ | $\le 4$ | $\le 2$ |
| | | | – | 2 | $\le 1$ | $\le 4$ | $\le 2$ | – |
| | | $(2,2)$ | 1 | – | – | $\le 4$ | $\le 6$ | $\le 3$ |
| | | | – | – | $\le 1$ | $\le 4$ | $\le 4$ | $\le 2$ |
| | $S^+ \to t\bar{b}$ | $(1,1)$ | 1 | – | – | $\le 6$ | $\le 4$ | $\le 2$ |
| $T\bar{T} \to S^+ W^- b\bar{b}$ | | | – | – | $\le 6$ | $\le 6$ | $\le 2$ | – |
| $T\bar{T} \to S^+ Z b\bar{t}$ | $S^+ \to$ bosonic | $(1,1)$ | 2 | – | – | $\le 6$ | $\le 6$ | $\le 3$ |
| $T\bar{T} \to S^+ H b\bar{t}$ | | | 1 | $\le 1$ | $\le 1$ | $\le 6$ | $\le 4$ | $\le 2$ |
| | | | – | $\le 1$ | $\le 1$ | $\le 4$ | $\le 2$ | – |

**Table 7:** Categorisation of final states for pair production of a vector-like top quark, where the bosonic decays of the BSM scalars are same as shown in Tab. 6.

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
