# Peer review of "Vector-like quarks: status and new directions at the LHC"

_SciPost Physics_

## Round 1 · Referee Report · Anonymous (Referee 1) · 2024-8-7

Strengths
Report
In Composite Higgs-like models, there are exotic bosons to which vector-like quarks can decay. These new decay modes are yet to be experimentally explored at the LHC. This paper considers this possibility in some detail. First, the authors theoretically motivate their study mainly in the context of CHMs and list the possible VLQs and pNGBs and the relevant new operators. Then, they discuss the pair and single productions of VLQs at the LHC, emphasizing the importance of looking beyond the NWA, as the VLQ widths can be considerable if the couplings involved in their decays are not small. They estimate the cross-sections of the VLQ production processes, retaining the interference with the corresponding SM processes to show that both large width and interference effects can be important for heavy VLQs in both pair and single production processes. In the next section, they consider an illustrative scenario of a singlet T quark with an exotic decay mode to a singlet scalar S0 and show how the new mode affects the cross-sections of the processes with standard and exotic final states. They summarize the Run 2 bounds on the VLQs from both pair and single production searches and sketch out the possibilities for Run 3 and HL-LHC.
Their paper is a nice and timely status update on the VLQ searches at the LHC. It is clearly written in a concise manner with experiment-friendly parameterization and enough details for the reader. Apart from the lack of rigorous simulations (i.e., including detector effects, etc.), the coverage is reasonably comprehensive. I recommend it for publication in SciPost Physics. However, before publication, the authors should address the following minor points.
- In the single production diagram in Fig.1, shouldn't the final heavy fermion be marked as (bottom/top) like the VB is written as W/Z?
- On page 5, the paragraph starting with "The effects of": m_t \lesssim 2 TeV should be m_T \lesssim 2 TeV.
- In Figs. 2 & 5, the negative numbers and the zeros on the Y-axes are a little confusing, as they are shown in log scales.
- The decays of the new bosons are briefly mentioned in the comment before Section 4 and in the appendix. Since we see the effects of the widths of VLQs, a comment on the widths of the singlet bosons might be interesting.
Recommendation
Publish (easily meets expectations and criteria for this Journal; among top 50%)

---

## Round 1 · Referee Report · Anonymous (Referee 2) · 2024-8-20

Strengths
- Comprehensive review of strategies to constrain vectorlike quarks in minimal scenarios (only couplings to SM particles) and in scenarios with an extra scalar particle.
- Extensive summary tables and numerical results. Useful as reference for both experimental and theoretical studies.
- Discussion of the impact of the vectorlike width.
Weaknesses
Report
Overall this paper is well written, contains a large amount of information and is going to be certainly of great aid to experimentalists and theorists alike.
Before I can recommend it for publication I would like to draw the attention of the authors on two points which I detail in the "Requested changes" section.
Requested changes
1.
It is not immediately clear how the cross sections for the irreducible backgrounds in table 3 are relevant. Experimental analyses, depending on the channel, require quite stringent cuts on various variables (pT, HT, missing energy, various invariant masses, ...) while the results in table 3 are presented with very loose cuts. In particular I was confused by the very small dR cuts. First of all there seems not to be any final state with two jets (but in footnote 4 a dRjj > 0.01 is mentioned). Second, in the limit in which mb is much smaller than the typical pT of final state particles, some of those cross sections have IR divergences at dR -> 0.
2.
In section 3 above eq (12a) and then in the first bullet point at page 9, it is stressed that the singlet T cannot be produced via S0. Because of this, the cross sections in figure 5 drop to zero for vanishing BR(T->S0 t). But, as mentioned above eq (12a), S0 has to couple to SM fermions/gauge bosons. This implies that one could have relatively strong production via gluon fusion to a heavy quark loop which couples to S0 with subsequent decay S0 -> T t: gg -> S0 -> Tt . This process is loop induced, is not suppressed by EW couplings but depends on the details of the couplings of S0 to SM particles (for instance, in figure 6, the couplings S0->bb, S->tt and others are considered). The authors should comment on this point. In particular it would be good to provide an understanding of the max value of BR(T->S0 t) above which the curves in figure 5 aren't accurate anymore and comment on whether the maximum at BR(T->S0 t) = 1/2 is still there if production via S0 is included.
Recommendation
Ask for minor revision

---

## Editorial Decision

unknown